# Phage Adsorption to Gram-Positive Bacteria

**DOI:** 10.3390/v15010196

**Published:** 2023-01-10

**Authors:** Audrey Leprince, Jacques Mahillon

**Affiliations:** Laboratory of Food and Environmental Microbiology, Earth and Life Institute, UCLouvain, 1348 Louvain-la-Neuve, Belgium

**Keywords:** phage, receptor, receptor binding protein, adsorption, tail, Gram-positive host

## Abstract

The phage life cycle is a multi-stage process initiated by the recognition and attachment of the virus to its bacterial host. This adsorption step depends on the specific interaction between bacterial structures acting as receptors and viral proteins called Receptor Binding Proteins (RBP). The adsorption process is essential as it is the first determinant of phage host range and a sine qua non condition for the subsequent conduct of the life cycle. In phages belonging to the *Caudoviricetes* class, the capsid is attached to a tail, which is the central player in the adsorption as it comprises the RBP and accessory proteins facilitating phage binding and cell wall penetration prior to genome injection. The nature of the viral proteins involved in host adhesion not only depends on the phage morphology (i.e., myovirus, siphovirus, or podovirus) but also the targeted host. Here, we give an overview of the adsorption process and compile the available information on the type of receptors that can be recognized and the viral proteins taking part in the process, with the primary focus on phages infecting Gram-positive bacteria.

## 1. Introduction

Bacteriophages (aka phages) are ubiquitous and are found in every place where their hosts, bacteria or archaea, are present. They are obligate parasites as they rely on their host replication and translation machinery to produce new virions. Phages are generally highly specific and often infect a single bacterial species or even only a few strains. Despite their small genome size, they are extremely diverse regarding their morphology and genome and also their lifestyle [1]. While virulent phages perform lytic cycles and are recognized as bacteria killers, temperate phages can perform lysogenic cycles in which they reside harmlessly within their hosts as prophages. Regardless of the life cycle, the first step always involves the recognition of the host and the attachment of the phage to the cell surface, a process common to all viruses infecting hosts from the three domains of life and described as the adsorption stage.

The phage adsorption to its host relies on the highly specific interaction between viral proteins called Receptor Binding Proteins (RBP) and bacterial receptors. The adsorption is generally depicted as a three-step process (Figure 1) [2]. First, phages encounter their hosts through random collisions due to diffusion in the medium and Brownian motion. In the second phase, a reversible interaction between RBP and cell surface components (i.e., the primary receptor) occurs and allows phages to remain near the host surface to facilitate the search for a secondary receptor to which they finally bind irreversibly. The adsorption can be assisted by depolymerases that enzymatically cleave cell wall (CW) polysaccharides to facilitate access to the receptors. Other viral proteins, called ectolysins, may be involved in the local degradation of the peptidoglycan (PG) layer to further facilitate genome injection. In phages belonging to the *Caudoviricetes* class, RBP are located at the distal part of the tail and are baseplate proteins, tail fibers, or spikes (Figure 1) [3]. As for the bacterial receptor, it can be any component exposed at the cell surface, and thus varies between Gram-negative and Gram-positive bacteria (Table 1) [2]. Apart from its nature, the receptor localization, accessibility, and density greatly influence phage adsorption. Interestingly, receptors and RBP involved in the reversible or irreversible adsorptions may be identical or different. For instance, phage LL-H infecting *Lactobacillus delbrueckii* recognizes different parts of lipoteichoic acids (LTA) for the reversible and irreversible steps using its tail spike, whereas *Escherichia coli* phage T5 first binds to lipopolysaccharides (LPS) using L-shaped fibers and then irreversibly attaches to the FhuA protein thanks to its tail protein pb5 [4,5,6].

Importantly, the adsorption step is the primary stage involved in resistance mechanisms developed by bacteria to prevent phage infection [7,8]. Indeed, bacteria can easily block receptor recognition either by receptor mutation or loss, or by producing an extracellular matrix that impedes its access [9,10,11]. In addition, prophages can express superinfection exclusion proteins that prevent secondary infection by other phages notably by blocking the RBP binding to the receptor [12]. Another strategy involves the stochastic expression of distinct receptors (i.e., expression linked to phase variation or growth phase) or quorum sensing regulated expression of the receptors [13,14,15,16,17]. Phages have however evolved counterstrategies to bypass these antiviral systems [18]: mutation of their RBP gene to recognize the novel receptor or adaptation of their polysaccharides degrading enzymes. This constant arms race between phages and their hosts forces us to further investigate the mechanisms underlying phage recognition and adsorption to ensure successful applications of phages in the medical sector and food industry.

The present review focuses on the phage adsorption process and describes phage RBP and their cognate bacterial receptors. Detailed information on the kinetics of phage adsorption can be found in [19]. Baseplate proteins with an accessory role for facilitating receptor attachment (i.e., depolymerases) and genome injection (i.e., ectolysins) are also briefly described, but more comprehensive reviews on these proteins can be found in [20,21]. Although the primary focus of this review is on phages infecting Gram-positive bacteria, a few iconic Gram-negative phages are also included for the sake of comparison.

**Table 1 viruses-15-00196-t001:** Example of receptors recognized by phages infecting Gram-positive and Gram-negative bacteria. The phage morphology is indicated: myovirus (M), siphovirus (S), or podovirus (P). WTA: wall teichoic acid, LTA: lipoteichoic acid, CW: cell wall, PG: peptidoglycan, LPS: lipopolysaccharide, Rha: rhamnose, Glu: glucose, GlcNac: *N*-acetyl-glucosamine, GalNac: *N*-acetyl-galactosamine, RboP: polyribitol phosphate, GroP: polyglycerol phosphate, Hep: heptulose.

Phage	Morpho	Host	Receptor	Reference
Gram-positive hosts				
Gamma	S	*Bacillus anthracis*	Membrane surface-anchored protein (GamR)	[22]
PBP1	S	*Bacillus pumilus*	Flagella (reversible)	[23]
SPP1	S	*Bacillus subtilis*	Glc residues on WTA (reversible) and membrane protein YueB (irreversible)	[24]
PBS1	S	*B. subtilis*	Flagella	[25]
CPS1	P	*Clostridium perfringens*	Capsular polysaccharides (glucosamine and galactosamine)	[26]
VPE25	S	*Enterococcus faecalis*	Membrane protein (PIP_EF_) as secondary receptor	[27]
ΦNPV1	S	*E. faecalis*	Exopolysaccharides as primary receptor	[28]
LL-H	S	*Lactobacillus delbrueckii*	Glu moiety of LTA (reversible) and negatively charged GroP group of the LTA (irreversible)	[4]
B1	S	*Lactobacillus plantarum*	Galactose of CW polysaccharides	[29]
c2	S	*Lactobacillus lactis*	Rha in PG (reversible) and membrane protein (PIP) (irreversible)	[30]
p2	S	*L. lactis*	CW pellicle	[31]
CHPC971	S	*L. lactis*	CW pellicle	[32]
A511	M	*Listeria monocytogenes*	GlcNAc and Rha on WTA and PG	[33]
A118	S	*L. monocytogenes*	Rha on WTA	[34]
P35	S	*L. monocytogenes*	GlcNAc and Rha on WTA	[35]
Φ11	S	*Staphylococcus aureus*	GlcNAc residue in RboP WTA, *O*-acetylated PG	[36]
187	S	*S. aureus*	GalNAc residue in GroP WTA	[37]
P68	P	*S. aureus*	β-GlcNAc residue in RboP WTA	[38]
SA012	M	*S. aureus*	α-GlcNAc residue and backbone of RboP WTA	[39]
SA039	M	*S. aureus*	β-GlcNAc residue and backbone of RboP WTA	[9]
K	M	*S. aureus*	Backbone of WTA	[36]
Dp-1	S	*Streptococcus pneumoniae*	Choline containing teichoic acids	[37]
9871	S	*Streptococcus thermophilus*	Exopolysaccharides	[40]
CHPC951	S	*S. thermophilus*	Rha-Glc CW polysaccharides	[41]
Gram-negative				
7-7-1	M	*Agrobacterium* sp.	Flagella and LPS	[42]
F336	M	*Campylobacter jejuni*	Capsular polysaccharides	[43]
φCb13	S	*Caulobacter crescentus*	Flagellum and pilus portals on the cell pole	[44]
T4	M	*Escherichia coli*	OmpC and LPS	[45]
T5	S	*E. coli*	Polymannose sequence in LPS and FhuA	[5,6]
λ	S	*E. coli*	LamB protein	[46]
SRD2021	S	*Klebsiella pneumoniae*	Capsular polysaccharide	[47]
MPK7	P	*Pseudomonas aeruginosa*	Type IV pili	[48]
JG004	M	*P. aeruginosa*	LPS	[49]
P22	P	*Salmonella enterica*	α-Rhamnosyl 1–3 galactose linkage of LPS *O*-chain	[50]
iEPS5	S	*S. enterica*	Flagella molecular ruler protein FliK	[51]
SPC35	S	*S. enterica*	BtuB protein	[52]
L-413C	M	*Yersinia pestis*	Terminal GlcNAc LPS outer core. HepII/HepIII and HepI/Glc residues also involved	[53]
YepE2	P	*Y. pestis*	HepII/HepIII LPS inner core	[54]

## 2. The Bacterial Side

The bacterial cell envelope is a complex network that comprises all the structures surrounding the cytoplasm, namely the inner membrane (IM) and the CW. It protects bacteria from their external environment while allowing the selective intake of nutrients and release of waste. The IM is a symmetric bilayer of phospholipids with embedded or anchored lipoproteins, glycolipids, and protein complexes. It is involved in essential processes such as energy production, protein secretion, lipid biosynthesis, and transport [54]. The layers outside the IM form the CW, which is notably responsible for maintaining the shape and integrity of the cell. One can distinguish two main types of bacteria based on the CW structure. Gram-positive bacteria possess a thick PG (i.e., 30–100 nm) associated with proteins. In addition, glycopolymers are abundant and either attached to the PG (e.g., wall teichoic acid) or anchored in the IM (i.e., lipoteichoic acids) [55]. Gram-negative bacteria have the particularity to possess a second membrane, the outer membrane (OM), which together with the IM delimits a compartment, called the periplasm, enclosing a thin PG layer. Some bacteria can also harbor additional protective layers such as Surface-layers (aka S-layers) and glycocalyx (capsules or slime layers).

### 2.1. The Gram-Negative Cell Surface

The OM is a lipid bilayer with phospholipids on the inner side and glycolipids, mostly lipopolysaccharides (LPS), on the outer side. LPS consist of three parts: a lipid A, a polysaccharide core, and the *O*-antigen. The lipid A is a glucosamine disaccharide with six or seven acyl chains that are embedded in the OM. The rest of the LPS molecule is a polysaccharide with a conserved central core and a terminal highly variable *O*-antigen. Because of their surface location, LPS are the most common receptors targeted by phages infecting Gram-negative hosts. They can either recognize the variable *O*-antigen (e.g., *E. coli* phage T5) or the conserved core polysaccharide (e.g., *Yersinia pestis* phage PST) [53,56]. Due to the variable nature of the *O*-antigen, phages recognizing this part are highly specific, whereas those binding to the LPS core usually have a broader host range [57]. However, phages such as SSU5 infecting *Salmonella* spp. and recognizing the core can only infect rough strains suggesting that the presence of the *O*-antigen can somehow hinder adsorption [58]. Interestingly, podoviruses targeting the *O*-antigen, such as *Salmonella* Typhimurium phage P22, possess depolymerases, which, upon adsorption, degrade the sugar moieties to facilitate the tail penetration (see Section 3.1.3) [59].

Proteins are other important components of the OM. Although the LPS create a physical barrier, the presence of outer membrane proteins (OMP), such as porins, allows the passage of small molecules [54]. Porins are integral proteins that adopt a beta-barrel conformation and span the entire OM. Some porins (e.g., OmpF and OmpC) allow the passive diffusion of monosaccharides, disaccharides, and amino acids, while others are very specific, such as LamB involved in maltose uptake. OMP are commonly identified as receptors in Gram-negative bacteria. For instance, phage λ infecting *E. coli* binds to LamB, whereas *E. coli* phage T5 adsorbs to the ferrichrome-iron transporter, FhuA, in addition to LPS [5,6,47]. The efflux pump TolC has also been identified as a receptor for phages infecting *S. enterica* serovar Typhimurium [60].

### 2.2. The Gram-Positive Cell Surface

#### 2.2.1. The Peptidoglycan Layer and Associated Proteins

The main function of PG is to maintain cell integrity, but it can also serve as an attachment site for proteins and polysaccharides [61]. The PG structural unit consists of a disaccharide of *N*-acetylglucosamine (GlcNac) and *N*-acetylmuramic acid (MurNac) linked by a glycosidic bond with a tetrapeptide attached to the MurNac residue. Multiple repeats of this unit form linear glycan chains that are cross-linked either through a direct covalent bond (most Gram-negative) or a peptide interbridge (most Gram-positive) between the stem peptides. The PG structure is highly conserved and similar between Gram-positive and Gram-negative bacteria. However, some variations exist and mostly occur in the composition and position of the crosslinking but also in the peptide stem composition or by the presence of secondary modifications in the glycan strands (e.g., *N*-deacetylation and *O*-acetylation) [60]. Contrary to Gram-negative bacteria, PG is directly exposed at the cell surface in Gram-positive bacteria and can, therefore, serve as a phage receptor. For instance, *Clostridium botulinum* phage α2 requires both MurNac and GlcNac residues for its adsorption [62]. However, due to the presence of these sugars in all bacterial CW, additional components such as the variable peptide bridges are thought to be necessary. Indeed, the PG often acts alongside another receptor as for *Listeria monocytogenes* A511 and *Staphylococcus aureus* ϕ11 phages, which are also relying on cell wall teichoic acids (WTA) [33,35,63].

In Gram-positive bacteria, surface proteins are mainly associated with PG either via ionic interactions or by covalent attachment. Proteins that are attached covalently possess an N-terminal signal sequence involved in protein export and a C-terminal pentapeptide motif (LPXTG) required for the sortase to catalyze the transpeptidation reaction between the sorting motif and the stem peptide [54]. Few protein receptors have been identified in Gram-positive bacteria. The *Bacillus anthracis* phage γ interacts with a surface protein, whereas *B. subtilis* phage SPP1, some *Enterococcus faecalis* phages, and a few lactococcal phages bind to membrane proteins that possess a large segment exposed at the cell surface [22,27,30,64].

#### 2.2.2. Gram-Positive Secondary Cell Wall Polysaccharides

Besides PG, the Gram-positive CW comprises other important glycopolymers that can be grouped under the general denomination of “secondary cell wall polysaccharides” (SCWP) [55]. Contrary to the PG, SCWP are highly variable and specific to bacterial species or even strains. They harbor various sugars (from trioses to hexoses) and differences in net charge and decorations (Figure 2). The SCWP can be divided into two main classes: the “classical SCWP” gathering teichoic and teichuronic acids and “non-classical SCWP”, structurally distinct and found in bacilli and relatives [65]. The exact roles of SCWP are still unclear but there is evidence of their implication in CW maintenance and turnover [66,67]. They also seem to be involved in biofilm formation and attachment to host cells (e.g., *S. aureus* WTA bind to epithelial cells) [68]. Moreover, SCWP can bind non-covalently to host proteins, notably to autolysins and choline-binding proteins in the case of *Streptococcus pneumoniae* [69].

Classical SCWP are highly abundant in Gram-positive CW and can account for up to 60% of their mass [66,67]. Teichoic acids (TA) are the main classical SCWP and can be of two types: either wall teichoic acid (WTA) or lipoteichoic acid (LTA). WTA are linked to PG, extend beyond the CW, and are composed of a conserved disaccharide unit (i.e., MurNac-GlcNac with one to two glycerol 3-phosphate) attached to the MurNac residues. The most frequently encountered WTA backbones involve either polyribitol phosphate (RboP) or polyglycerol phosphate (GroP) chains, with 40–60 repeats [66]. Both WTA backbones can be found within the same species (e.g., *B. subtilis* and *S. aureus*) while some species completely lack WTA (e.g., *B. anthracis* and *Streptococcus thermophilus*) [73,74]. RboP and GroP repeats can be tailored with d-alanine esters and a variety of mono-or oligosaccharides [75]. Regarding LTA, they are GroP polymers that are anchored to the IM [66]. They are also functionalized with d-alanine or sugars but contain fewer GroP repeats so they do not extend beyond the PG contrary to WTA. LTA are less diverse than WTA, although complex structures have been identified in *S. pneumoniae* [55]. Finally, a second class of classical SCWP, called the teichuronic acids, is found in bacilli growing in phosphate-limited conditions. In these bacteria, TA synthesis is replaced by that of teichuronic acid, which contains uronic acids [76].

Non-classical SCWP are found in various bacilli (e.g., *Paenibacillus alvei*, *Bacillus cereus*, *B. anthracis,* and *Geobacillus tepidamans*) and can account for 7 to 15% of the CW weight [65]. They contain 2–15 repeats of mostly linear di-, tri-, or tetra-saccharide strands that are rich in hexosamine and hexosaminuronic acids and can harbor non-carbohydrate modifications such as pyruvate, phosphate, and acetate. One single type of SCWP is found per organism. In S-layer-producing species, such as *B. anthracis*, SCWP serve as attachment sites of S-Layer proteins through non-covalent interactions [73].

Unsurprisingly, WTA are the most widespread receptors targeted by phages infecting Gram-positive hosts [77]. They are notably involved in the adsorption of phages infecting *Staphylococcus* spp., *Listeria* spp., and *B. subtilis*. Similarly, LTA have also been identified as receptors, as in the case of phage LL-H infecting *L. delbrueckii* [4]. This phage reversibly binds to the glucose moieties on LTA and then irreversibly attaches to the negatively charged GroP. In *Lactococcus lactis*, particular polysaccharides are implicated in phage adsorption [32]. These polysaccharides seem to be covalently bound to another SCWP containing rhamnose and form a compact pellicle located on the outer surface of the bacteria [78]. They are, however, distinct from capsules (see below) and other SCWP identified in Gram-positive bacteria to date [79].

### 2.3. Other Associated Envelope Components

#### 2.3.1. S-Layer 

The S-layer is an outermost envelope component composed of identical proteins or glycoproteins that are organized as a two-dimensional crystalline array forming pores of 2–10 nm [73]. It is maintained on the cell surface via non-covalent interactions with LPS or SCWP. The S-layer functions as a protective barrier against environmental stresses (e.g., resistance to low pH). It is also involved in bacterial adhesion and can participate in the maintenance of cell shape [80]. The small pore size can also prevent phage infection. Plaut and Coll [81] showed that deletion of the *sap* gene encoding the S-layer protein in *B. anthracis* prevents the adsorption of phage AP50c, thus highlighting the involvement of the S-layer in this phage adsorption.

#### 2.3.2. Capsules and Exopolysaccharides (EPS)

Another structure that can surround the CW is a capsule that is generally made of polysaccharides tightly bound to the PG. In rare cases, it is constituted of proteins (e.g., the poly d-glutamic acid capsule in *B. anthracis*). The capsule’s primary role is to protect against phagocytosis and desiccation, but it is also involved in bacterial adhesion and virulence. The capsule may act as an obstacle to phage adsorption by masking the receptor in the CW [8]. However, some phages can adsorb to capsular polysaccharides. This adsorption is generally reversible and involves the enzymatic degradation of capsular polysaccharides by depolymerases to gain access to the true receptor located in the CW. Phages targeting capsules have notably been identified in *E. coli*, *Klebsiella pneumoniae*, *S. enterica*, and *Acinetobacter baumannii* [10,82,83,84]. Interestingly, in *B. subtilis* subsp. *natto*, some cells are devoid of capsules and can be infected by the ϕNIT1 phage, which produces depolymerases released during the lysis step. These enzymes, in turn, allow the breakdown of capsules of neighboring encapsulated cells, thus allowing the phage to spread [85].

Bacteria can also produce another type of polysaccharide, EPS, which contrary to capsular polysaccharides is loosely attached to the cell wall (i.e., Slime layer) or directly secreted in the extracellular environment [86,87]. EPS have the same protective role as bacterial capsules but in addition, they are a major component of the biofilm matrix [88]. EPS was shown to serve as the primary receptor for the *E. faecalis* phage NPV1 [28]. Similarly, EPS structures are essential for the adsorption of *S. thermophilus* phages from the 987-group [40].

#### 2.3.3. Cell Appendages

Flagella and pili are surface appendages found in both Gram-negative and Gram-positive bacteria. Flagella are implicated in locomotion, whereas pili are involved in attachment to other cells or surfaces, and are implicated in the conjugation of Gram-negative bacteria. Adsorption to flagella seems to be an advantage in situations where the likelihood of encountering a host is low. Phage PBP1, infecting *Bacillus pumilus*, attach to flagella in the first reversible binding step using their tail tip [23]. Following this primary attachment, the flagellum rotation allows phages to move toward the cell surface where they interact irreversibly with a secondary receptor and inject their genome. Several *Pseudomonas* spp. phages recognize and adsorb to the type IV pili, and retraction of the pilus is thought to bring phages in close contact with the bacterial surface where they can irreversibly interact with the receptor [48,89,90].

## 3. The Phage Side

The phage tail is implicated in the recognition and attachment to the bacterial host and the correct positioning of the virion before DNA injection [3]. Its length and complexity depend on the host and the receptor recognized. The distal part of the tail usually forms a macromolecular structure called the baseplate, which comprises proteins directly involved in host interactions [91]. Among them, the RBP are the key players in the adsorption, as it is the interaction of these proteins with bacterial receptor(s) that trigger conformational changes leading to DNA injection into the bacterial cell. Not surprisingly, RBP evolve faster than the rest of the phage genome because changes in their structure may induce host range modifications that can be favorable [92,93].

### 3.1. Myoviruses

Myoviruses are characterized by a contractile tail, which is the most complex tail apparatus. The structure of *E. coli* phage T4 baseplate is the best characterized so far, although it appears to be really complex compared to those of other myoviruses (e.g., *E. coli* phage Mu) [94]. Despite the differences in terms of cell envelope between Gram-positive and Gram-negative bacteria, the tails of their respective myoviruses share many features, and *E. coli* phage T4 will be taken as a model here due to the abundance of information related to this phage [95]. The simplest tail structure encountered in myoviruses is depicted in Figure 3.

#### 3.1.1. The Tail Tube

The tail tube attaches the capsid on one side and the baseplate on the other and is formed by stacked hexameric rings of the tail tube protein or major tail protein (MTP) [96]. It encloses the tape measure protein (TMP) that defines the tail length and may harbor a C-terminal PG hydrolase domain involved in local PG degradation to facilitate CW penetration. The tail tube is surrounded by a contractile helical sheath, implicated in cell envelope penetration upon adsorption, and is capped at each side with terminator proteins (i.e., tail tube and tail sheath terminators) [97,98]. In phage T4, the neck (i.e., the intersection between the tail and the head) is decorated with 12 trimeric proteins; six of them form the collar and are folded around the neck, whereas the other six point downward and are called whiskers. These proteins are environmental sensors that influence the retraction or deployment of T4 tail fibers [98].

#### 3.1.2. The Baseplate

In myoviruses, the baseplate can be divided into two parts. First, the hub complex is the central part that assembles the other tail components. The first baseplate hub (BH) protein (i.e., gp48/gp54 in phage T4), also named the tail tube initiator, serves as a starting point for the tail tube polymerization. The second BH protein is conserved among myoviruses and siphoviruses and is the intermediate between the threefold symmetry of the baseplate hub and the sixfold symmetry of the rest of the baseplate [98]. This protein interacts with the tail spike protein to form a central cell-puncturing device involved in membrane penetration and peptidoglycan degradation upon phage adsorption. This cell-puncturing device is well characterized in phage T4 and formed by gp27 (BH protein), gp5 (spike), and the needle tip gp5.4. Gp5 contains a lysozyme domain and a C-terminal domain rich in β structures that are commonly found in membrane-puncturing proteins [96]. In phages infecting Gram-positive bacteria (e.g., *Listeria* phage A511), the membrane-interacting domain is rather formed by coiled-coil structures [99]. 

The second part of the baseplate is formed by six peripheric wedges whose role is to anchor the tail fibers (i.e., RBP) to the central hub [100]. The baseplate wedges are composed of at least three conserved proteins. The RBP are trimeric structures mostly composed of intertwined β-helix and less often of α-helical coiled coils [97]. The N-terminal domain anchors the protein to the baseplate, whereas the C-terminal part contains the receptor binding domain. RBP are either tail fibers or spike proteins [3]. Tail fibers are usually complex and require chaperones for their folding and trimerization, whereas spikes are simpler and usually possess an enzymatic activity [3,98]. In T4 for instance, two types of tail fibers are present (Figure 3) [96,101]. The long tail fibers (LTF) are composed of four distinct proteins (i.e., gp34 to gp37), with the receptor binding region located at the C-terminal part of gp37. The second type is short tail fibers (STF), which are homotrimers of gp12 that are folded beneath the baseplate. T4 LTF alternate between retracted and extended conformations. When a phage encounters a host, one or two extended LTF initiate contact with the LPS or OmpC [101]. This interaction permits the phage motion along the surface in order to find a suitable infection site. Ultimately, the other LTF bind to the surface leading to the perpendicular orientation of the phage to the cell. This induces changes in STF conformation that rotate to point downwards, thus allowing the irreversible binding to the outer core of the LPS [3]. 

#### 3.1.3. Tail Proteins Associated with Enzymatic Activity

Phages often require breaking down some cell envelope structures, either to access the receptors or to facilitate DNA injection following adsorption [20]. Therefore, some tail proteins harbor different enzymatic activities to achieve these goals. These proteins are either depolymerases, targeting polysaccharides, or ectolysins, breaking down the PG.

As indicated earlier, some depolymerases break down the capsule surrounding bacteria that hamper adsorption by masking the receptor such as the one found in *K. pneumoniae*, *E. coli,* or *A. baumannii* [102,103]. In phages recognizing *O*-antigen or capsules, the RBP possess depolymerase activity to specifically cleave the saccharide repeating units and bring the phage closer to the bacterial surface [104,105]. Furthermore, phages with depolymerase activities have a great advantage in infecting bacteria embedded in biofilms [106]. Depolymerases have mainly been characterized in phages infecting Gram-negative bacteria, but experimental evidence of depolymerase activity for phages infecting Gram-positive hosts is available for *B. subtilis* phage phi29 (i.e., tail spike with activity against WTA) and *S. aureus* phage vB_SepiS-phiIPLA7 (i.e., depolymerase targeting EPS) [107,108]. Depending on their enzymatic activity, depolymerases have been divided into two classes: hydrolases (e.g., sialidases and rhamnosidases) and lyases (e.g., hyaluronidases and alginases) [109]. Although phage depolymerases are mostly tail components (i.e., fibers and spikes), some have been identified as neck proteins or proteins synthesized during host lysis [109]. Interestingly, some phages harbor distinct tail proteins with different depolymerase activities (e.g., *K. pneumoniae* phage KP32) [110].

Ectolysins (or virion-associated lysins) are PG-degrading enzymes that are involved in the degradation of the CW following phage adsorption [20]. These proteins locally degrade PG without harming the cell integrity to facilitate the tail tube penetration through the rigid CW. However, when phages are applied at high multiplicity of infection, an ectolysin-mediated lysis can occur as a result of the multiple “holes” formed in the CW. This process, called “lysis-from-without”, leads to bacterial lysis without phage multiplication [111]. The functional domains found in ectolysins have either glycosidase or endopeptidase activities. Interestingly, ectolysins from phages infecting Gram-negative bacteria feature a single catalytic domain, whereas those found in Gram-positive phages frequently have two different catalytic domains, which is probably linked to the thicker PG layer [112]. Ectolysins are primarily associated with tail proteins or TMP, but a few have been identified as neck proteins or capsid inner proteins that are ejected prior to DNA [113]. Ectolysins are not present in all phages and have sometimes been found to be dispensable for phage infection. Instead, they rather confer an advantage in certain physiological conditions where the PG meshwork is highly cross-linked (i.e., in the stationary phase) [113].

### 3.2. Siphoviruses

In siphoviruses, the tail morphogenesis module is usually encoded between the TMP and the lysis cassette (i.e., holin and endolysin genes) with the exception of the MTP gene, which is located upstream tmp (Figure 4A) [77]. The Distal tail (Dit) protein and the tail lysin (Tal) are encoded directly downstream *tmp* and are always present in siphoviruses. Together, these three proteins form the initiator complex onto which RBP, auxiliary proteins, and MTP are assembled to form the complete tail structure that attaches the capsid (Figure 4B) [114]. 

#### 3.2.1. The Tail Tube

The TMP interacts with the tail terminator on its N-terminal end and with the Dit hexamer and Tal on its C-terminal part [115]. As for myoviruses, the MTP polymerized around the TMP to form the tail tube. Interestingly, MTP can exhibit decorations such as adhesins (e.g., p2 and λ phages) or complete Carbohydrate Binding Module (CBM) (e.g., lactococcal phages) that are involved in the first reversible binding or increase the phage adsorption to its host [31,116]. Similarly, the Neck Passage Structure (NPS) found in some phages may harbor CBM that assists phage adsorption.

#### 3.2.2. Baseplate Proteins

The Dit proteins form a hexameric structure onto which the other tail components are attached (Figure 5). The N-terminal parts of the Dit monomers interact with each other and delimit a central channel, whereas the C-terminal parts are not connected and possess a galectin-like fold (i.e., phages SPP1 and TP901-1) or an oligosaccharide/oligonucleotide-binding (OB) fold (i.e., T5) [91,117,118]. Phages, such as p2, that directly assemble the RBP onto the Dit hexamers, harbor a long extension (“arm”) within the C-terminal domain, which is involved in the anchorage of the RBP [119]. On the contrary, this extension is missing in phages that do not assemble peripheric RBP (i.e., SPP1) (Figure 5) or rely on the attachment of auxiliary proteins interacting with the RBP (i.e., TP901-1) [91,118]. In phage p2, a second Dit hexamer is part of the baseplate and is attached to the first Dit hexamer through its N-terminal domain ring (Figure 6B) [120]. Interestingly, particular Dit proteins, called evolved Dit (evoDit) proteins, have been identified in several phages. Besides their architectural role, these proteins are also implicated in phage adsorption as they harbor CBM able to bind to the host polysaccharides [121,122]. By being more exposed, evoDit proteins are thought to be involved in the first reversible binding, probably to increase the possibilities of host attachment in any virion orientations [116]. The involvement of evoDit proteins in adsorption has only been shown for a few phages mostly infecting *Lactobacillus* spp., *L. lactis,* and *B. cereus*, but in silico analysis revealed that they are widespread in siphoviruses including those infecting *Streptococcus* spp., *L. monocytogenes*, or *Mycobacterium* spp. [112,116,123].

Regarding the Tal, it forms a trimer located at the tail extremity and closes the central channel formed by the tail tube and the Dit hexamer. Although the Tal N-terminal part is always involved in the interaction with the Dit central hub and the TMP, the rest of the protein exhibits a high structural and functional diversity. In phages binding to proteinaceous receptors, the Tal anchors the RBP (e.g., T5) or contains the receptor binding domain and is in fact the *bona fide* RBP (e.g., SPP1 and λ). The Tal can also harbor CBM and improves the adsorption of the RBP [124]. In addition, Tal may also display a PG-degrading activity assisting the phage in local PG break down upon adsorption in order to facilitate genome injection [125]. Of note, some phages, such as p2, in which the Tal is devoid of PG-degrading activity, are not able to infect bacteria in the stationary phase where the PG meshwork is highly cross-linked [31].

RBP form homotrimers in which their N-terminal part is responsible for the attachment to the baseplate and the C-terminal part is involved in host recognition, both domains being separated by a linker region. The modular organization of RBP allows the exchange of receptor binding domains, thus permitting host range adaptation [126]. The RBP structure of several phages infecting *L. lactis* (e.g., p2, TP901-1, and 1358), *L. monocytogenes* (e.g., PSA), and *S. aureus* (e.g., Φ11 and Φ80) are described in detail below [92,127,128,129,130,131]. In addition to the RBP present in the central tail spike, phages λ and T5 also harbor LTF involved in the reversible binding to OmpC and *O*-antigen, respectively [132,133]. 

Besides the above-mentioned proteins, siphophage baseplates can harbor other axillary proteins. In TP901-1, each Dit monomer anchors three trimers made of the Upper Baseplate proteins (BppU), which in turn attach three copies of the RBP trimer, thus forming a tripod structure (Figure 6) [91,134]. A similar protein (i.e., p132) is found in phage T5 and is involved in the anchorage of the LTF (Figure 4). Phage Tuc2009 displays a similar baseplate organization to that of TP901-1, except that an Accessory Baseplate protein (BppA) is present in the tripod [135]. This protein interacts with the N-terminal domain of BppU and harbors a CBM able to bind to the host, in a similar way to that of evoDit proteins [136].

#### 3.2.3. Baseplate Architecture

The number of RBP copies varies from one phage to another. For instance, in phage p2, one RBP trimer is connected to each Dit protein resulting in 18 copies of RBP monomers, whereas in TP901-1, three RBP trimers are attached to each BppU timers, yielding a total of 54 RBP copies (Figure 6). The presence of such a high number of receptor binding sites compensates for the weaker affinity for polysaccharide receptors compared to protein receptors. Indeed, phages recognizing protein receptors usually harbor a simpler baseplate with a single tail spike as RBP (e.g., SPP1). Interestingly, some phages (e.g., p2) can have two different baseplate conformations (Figure 6). In the rest conformation, the RBP head domains point upward, toward the capsid, which is not optimal for receptor recognition. However, in the presence of Ca^2+^, the baseplate switches to an activated state in which RBP rotate 200° to point downward, and the Tal trimer opens to form a passage large enough to allow the DNA transition [31,119]. On the contrary, some phages (e.g., TP901-1, SPP1, and T5) have a baseplate in a “ready-to-adsorb” conformation where the RBP point downward and do not require activation for efficient infection (Figure 6) [91].

### 3.3. Podoviruses

Podoviruses have short non-contractile and non-flexible tails. Contrary to myoviruses and siphoviruses, they do not possess any baseplate. *E. coli* phage T7 and *B. subtilis* ϕ29 are described in this section. 

Phage T7 tail begins with a dodecameric tail adaptor that binds to the portal complex to retain DNA within the capsid and connect the tail (Figure 7) [137]. A hexameric protein called the nozzle is attached to the adaptor, as well as six tail fibers pointing upwards and interacting with the capsid prior to infection [96]. Interestingly, some podoviruses are more complex and possess multiple tail fibers or spikes. For instance, the *E. coli* phage K1–5 has two types of spikes recognizing different capsular polysaccharides [138]. T7 first binds to a membrane protein before interacting with the inner core of the LPS. Conformational changes lead to the reorientation of the tail fibers and the opening of the nozzle [96]. The inner core proteins (i.e., gp14, gp15, and gp16) located in the capsid move near the portal ring and form a channel that extends the tail and spans the entire periplasm [139] (Figure 7). Among these proteins, gp16 has a lytic transglycosylase activity and breaks down the PG locally. The DNA is then translocated through the tail and the periplasmic tube and is delivered into the cytoplasm.

The podovirus ϕ29 infecting *B. subtilis* possesses a longer tail (about 50 versus 30 nm for T7), which may be related to the thicker CW of Gram-positive bacteria [96]. The virion structure is similar to that of T7 and P22, except that the tail tube is extended by a distal knob formed by a hexamer of gp9 and probably two copies of gp13 [140]. The latter possess an N-terminal lysozyme-like domain and a C-terminal endopeptidase activity that are involved in the CW penetration [141]. Gp13 is thought to form a pore in the IM to permit genome injection [142]. In ϕ29, twelve appendages or tail spikes are attached to the collar, two of them are in an “up conformation” while the rest are in “down conformation” [140]. Each appendage is a homotrimer of gp12 and possesses an intramolecular chaperone (IMC) domain that is absent in the final virion structure. The appendages recognize and digest the *B. subtilis* WTA. Interestingly, the capsid of ϕ29 is decorated with 55 fibers that are thought to enhance the phage adsorption to the CW [143].

## 4. Adsorption of Phages Infecting Gram-Positive Bacteria

The adsorption process of phages targeting Gram-positive bacteria is far less characterized than that of those infecting Gram-negative bacteria [3,56]. Yet, characterizing the molecular determinants involved in the adsorption of these phages is paramount as they are increasingly investigated as therapeutic alternatives for antibiotic-resistant bacteria such as *S. aureus* or because phage infection constitutes a major threat when bacteria, such as lactic acid bacteria, are used in industrial settings. 

### 4.1. Phages Infecting Lactic Acid Bacteria

Phage infection is an important issue in the dairy industry as prophage induction or virulent phage contamination can kill Lactic Acid Bacteria (LAB) used for fermentation [144]. To control phage propagation, many studies have focused on the adsorption process of phages infecting *L. lactis* and *S. thermophilus*, which are widely used in the industry.

#### 4.1.1. Lactococcal Phages

Phages targeting *L. lactis* display siphovirus or podovirus morphologies and have been further divided into ten groups based on morphological and genomic characteristics. The most encountered in the industry are siphoviruses belonging to the *Skunavirus* genus (formerly named the 936 group), the P335 group, and the *Ceduovirus* genus (formerly named the c2 group) [145].

*L. lactis* phages usually have narrow host spectra, targeting only a limited number of strains [146]. Those belonging to the *Skunavirus* (e.g., p2) and P335 group (e.g., TP901-1) recognize highly variable CWPS (Cell Wall Polysaccharides) forming the pellicle, which is a particular structure that covers the surface of *L. lactis* and is composed of repeating units of hexasaccharides linked by phosphodiester bonds [79,147,148]. The sugar composition of the pellicle varies from one strain to another, and *L. lactis* strains have been classfied into four groups (i.e., A-C and U) based on the composition of the gene cluster involved in the biosynthesis of these polysaccharides [149]. As a result, a given phage is only able to infect strains possessing the same type of CWPS or sometimes strains belonging to two different CWPS group types. Additionally, on analysis, the RBP C-terminal parts of phages infecting strains belonging to the 936 group were shown to cluster in three groups that correlate to the CWPS types they recognize [149]. CWPS are also recognized as receptors by siphophages belonging to four other groups (i.e., 949, P087, 1358, and 1706) that are rarely encountered in the industry [32,127,150]. By contrast, members of the *Ceduovirus* genus (e.g., c2) are the only lactococcal phages that base their adsorption on a proteinaceous receptor named Phage infecting protein (Pip) (or a structurally similar protein), which is an integral membrane protein with a portion exposed at the cell surface [30]. Nonetheless, phages from the *Ceduovirus* are thought to also rely on polysaccharides (i.e., rhamnose) for their initial reversible adsorption [151].

The baseplate structure of phages from the *Skunavirus* is highly conserved and similar to that of p2 (Figure 6B). However, Hayes and colleagues recently showed that out of the 115 available phage sequences of this group, three display a different gene organization, with the presence of a second RBP gene in Phi4.2, Phi4R15L, and Phi4R16L [136]. Of note, phages from the P335 group have a ready-to-adsorb baseplate contrary to that of phages from the *Skunavirus*, which is activable (Figure 6A,B). RBP structures of most lactococcal phages share the common three-domain organization with an N-terminal shoulder domain inserting the RBP in the baseplate, a central neck domain, and a head domain responsible for receptor binding (Figure 8) [119,130]. In phage 1358 (i.e., the representative of the rarely encountered 1358 group), the central neck domain is absent [127]. Regarding phages from the *Ceduovirus* genus, which rely on protein receptors, little is known about the viral proteins involved in adsorption.

#### 4.1.2. Streptococcal Phages

Phages infecting *S. thermophilus* are divided into five groups: the *pac* and *cos* groups (recently attributed the genus names *Brussowvirus* and *Moineauvirus*, respectively) comprising phages commonly encountered in the industry, and the 5093, 987, and P738 groups, for which only a few members have been isolated [154].

Similarly to phages targeting *L. lactis*, those infecting *S. thermophilus* recognize polysaccharides as receptors. Two types of CWPS are found in *S. thermophilus*. On one side, the Rhamnose-Glucose Polysaccharides (RGP) are highly branched rhamnose-rich glycans closely associated with the CW recognized by the *cos* and *pac* groups [41]. On the other side, exopolysaccharide (EPS) are more loosely associated and made of hexasaccharide repeating units with lactosyl side chains and are the target of phages from the 987 group [40].

Phages from the *pac* and *cos* groups have similar baseplate organization [124]. They all possess evoDit proteins, with one CBM insertion, whereas the Tal varies in length and complexity. Interestingly, Tal encoded by *cos* and *pac* phages all harbor one or two CBM resembling the BppA of TP901-1. It was shown that the Tal CBM are involved in adsorption although with lower binding affinity than the related RBP [124]. The bona fide RBP is encoded directly after the Tal gene and possesses a N-terminal BppU fold and a C-terminal domain similar to those of p2 and TP901-1 [124,155].

Phages of the 5093 group display tails with globular appendages attached to the tail tip [124]. They have classical Dit proteins and Tal but the RBP possesses an esterase fold in its C-terminal part. Phages belonging to the 987 group possess shorter tails than phages from the previous groups (~130 nm versus > 200 nm) [124]. They have classical Dit protein and their Tal harbor PG-degrading domains in the C-terminal part. The RBP appears to be a fusion between the BppU in N-ter and the RBP of the podovirus ϕ29 in its C-terminal part.

### 4.2. Listeria spp. Phages

Phages infecting *Listeria* spp. are mostly siphoviruses, although a few myoviruses have also been characterized [156]. In *Listeria* spp., 12 different serovars have been distinguished based on the substitutions found on their RboP WTA, and phages rely on these substitutions for their adsorption.

The siphoviruses P35 and A118 infect *L. monocytogenes* serovar 1/2, which is characterized by GlcNAc and Rha substitutions [34]. Although infecting the same serovar, both phages display many differences including lifestyle (P35 is virulent and A118 temperate) and adsorption strategies. P35 depends on both GlcNAc and Rha for its adsorption, whereas A118 only requires Rha substitutions. In P35, the protein gp16 was identified as the RBP while two proteins of A118 (i.e., gp19 and gp20) were able to bind to *Listeria* cells. The authors have suggested that one protein may be involved in the first reversible interaction and the other in the final irreversible attachment [34]. 

A511 is a myovirus infecting almost all strains of *L. monocytogenes* and *Listeria ivanovii* [33]. Its baseplate is similar to that of phage T4 and may constitute a model for the *Spounavirinae* subfamily (i.e., *Herelleviridae*) (Figure 9A) [99]. Although they infect different hosts, the central hub in A511 is similar to T4, suggesting that the baseplate spike gp99 is involved in IM penetration. The virion harbors six tail fibers composed of a proximal and a distal segment. The proximal part is made of three interacting dimers of the baseplate wedge protein gp105 to which are attached two trimers of gp106 via another baseplate protein (gp104). Gp106 trimers form pyramidal structures responsible for the double-layered baseplate observed in some myoviruses, as one trimer points toward the capsid while the other one points downward [33]. The distal part is formed by a trimer of gp108, the RBP that was shown to bind to the host surface and interact with GlcNAc and Rha substituents of the WTA (Figure 9B) [33]. Similarly to T4 STF, the RBP gp108 are folded and interact with the baseplate. Following this first binding, gp106 pyramids that were previously facing up reorient themselves to point toward the host and interact with the CW (Figure 9C) [99]. It is possible that another protein, gp107, may be part of the pyramids and be responsible for the binding. 

### 4.3. Staphylococcal Phages

Phages infecting *Staphylococcus* spp. are tailed phages classified into 11 groups (A–H and J–L) [157]. Apart from groups E, J, and K, which are specific to coagulase-negative staphylococci (CoNS) (e.g., *Staphylococcus epidermidis*), the vast majority of these phages infect *S. aureus* [158]. WTA appears to be the essential receptor for the adsorption of staphylococcal phages. In *S. aureus*, WTA are mostly present as repetitions of RboP substituted with GlcNAc and d-alanine. Two glycosyltransferases, TarM and TarS, are responsible for the addition of either α-GlcNAc and β-GlcNAc, respectively [159]. By contrast, CoNS produce WTA with a GroP backbone substituted with different types of sugars (e.g., GlcNAc, GalNAc, or Glc) [160].

It was shown that siphoviruses adsorb onto GlcNAc decorations of WTA independently of their stereochemistry as the adsorption was only completely abolished in the double knockout mutant (i.e., ΔtarMΔtarS) [161]. Similarly to CoNS, the *S. aureus* lineage ST395 harbors GroP WTA with α-GalNAc (glycosyltransferase TarN) and is not sensitive to the siphoviruses that infect most *S. aureus*. Instead, phages able to recognize this lineage (e.g., ϕ187) can also infect CoNS and recognized the GalNAc residue [36]. Interestingly, in addition to WTA, *O*-acetyl groups at the 6-position of MurNac residues in PG were also found to be implicated in the adsorption of the well-known siphovirus ϕ11, although to a lesser extent [35]. Regarding the baseplate of phage ϕ11, the protein gp45 was identified as the RBP, and the BppU gp54 seems to also be implicated in phage adsorption [35].

Gp45 does not exhibit structural homologies with other RBP, including those of Lactococcal phages that also rely on a saccharidic receptor. Instead, gp45 monomers can be divided into three domains: an N-terminal “stem” that is folded by 155° and attaches the RBP to the Dit hexamer, a central “platform” containing the binding site for GlcNAc, and a C-terminal “tower” domain containing two highly similar regions (Figure 10A). Six copies of homotrimeric RBP are found in ϕ11 baseplate, which are oriented downward in a ready-to-infect state [129]. Recently, high-resolution cryo-electron microscopy was used to determine the complete baseplate structure of the closely related staphylococcal siphovirus 80α (Figure 10B) [128]. In this phage, the RBP structure is highly similar to that of phage ϕ11 with the main difference being the orientation of the stem domain. In addition, phage 80α harbors two types of tail fibers (i.e., Lower and Upper tail fibers) that may be involved in adsorption although this remains to be proven. Similarly to phage ϕ11, 80α has a Tal with PG-degrading activity.

Unlike siphoviruses, most podoviruses require the presence of β-GlcNAc in WTA for infection while α-GlcNAc glycosylation prevents phage adsorption [37,162]. Nevertheless, Uchiyama and colleagues showed that phage S24-1 uses GlcNAc epitopes as receptors regardless of the glycosidic linkage [162]. The complete structure of the podovirus P68 was recently determined [163]. Interestingly, P68 possesses five trimeric head fibers, which, contrary to other phages, are located in the lower part of the capsid with the C-terminal part positioned at the same level as the receptor binding domain of the tail fibers. They seem to be involved in the right positioning of the virion for genome injection. Otherwise, the tail structure is similar to that of ϕ29 with a tail knob and a tail spike, the latter being involved in PG degradation. Twelve tail fibers are attached to the connector and display a similar three-domain organization to that of phage ϕ11 RBP.

Myoviruses, especially those belonging to the Kayvirus genus, display a wide host range and high lytic properties, which make them important candidates for the control of pathogenic *S. aureus* [70]. It was previously shown that myoviruses infecting *S. aureus* target the WTA backbone as receptors [164]. However, studies on phages ϕSA012 and ϕSA039 highlighted that they also rely on glycosylation for adsorption [9,38]. Their broad host ranges can be explained by the fact that these phages encode two types of RBP, one recognizing the WTA backbone and the other the GlcNAc decorations.

### 4.4. Bacillus spp. Phages

Very little is known about the adsorption process of phages infecting the genus *Bacillus*. The main data available concern the SPP1 and ϕ29 phages (see Section 3.3) infecting *B. subtilis*, as well as phage γ targeting *B. anthracis*. 

The receptor and viral proteins involved in phage SPP1 adsorption have been studied in detail [165]. First, SPP1 reversibly binds to glycosylated GroP WTA at the surface of *B. subtilis* [24]. In a second step, the phage adsorbs irreversibly to the membrane protein YueB, which is an orthologue of Pip, the membrane protein involved in the adsorption of *L. lactis* phages from the c2 group [64]. YueB is anchored in the IM by one N-terminal and five C-terminal TMD with a large central extracellular domain long enough to span the entire CW [166]. On the phage side, the tail spike gp21 (also called Tail Adaptor Protein: TAP) was identified as the RBP interacting with YueB [167]. A single trimer of gp21 is attached to the baseplate, which comprises two classical Dit hexamers [118]. The C-terminal domain of gp21 is responsible for the binding and subsequent triggering of genome injection, whereas the N-terminal domain forms a cap within the lower Dit hexamer, thus preventing the early release of DNA [167]. 

*B. anthracis*, the etiological agent of anthrax, belongs to the *B. cereus* group, a cluster of genetically closed bacteria with distinct virulence (e.g., the entomopathogen B. thuringiensis versus foodborne pathogenic strains of *B. cereus*) [168]. The siphophage γ is highly specific to *B. anthracis* and has been used for years in anthrax diagnosis [169]. Its receptor was identified as a CW-anchored protein called GamR [22]. This protein possesses an LPXTG motif that allows its attachment to the peptide moiety of PG thanks to sortases. In phage γ, two tail proteins are encoded between the TMP and the lysis cassette. Early characterization of the first tail protein, gp14 (YP_338197.1), revealed that it is involved in the phage γ adsorption and it was assumed that this protein was a tail fiber [170]. However, our recent analysis of the tail proteins of various siphoviruses infecting the *B. cereus* group (including those targeting *B. anthracis*) showed that gp14 possesses a central CBM and is in fact an evolved Dit [122]. As for the second tail protein encoded by phage γ, gp15, its implication in the adsorption process was not assessed but analysis of different γ isolates highlighted that the gene encoding this protein is a hot spot for mutations, consistent with its involvement in phage adsorption [170]. 

Regarding phages infecting other members of the *B. cereus* group, several studies showed that they probably rely on carbohydrates for their adsorption, but the exact nature of these receptors has never been demonstrated [171,172]. Recently, we performed an in silico analysis of the tail proteins found in siphoviruses infecting the *B. cereus* group [122]. Five different genetic organizations were identified in these phages with the most prevalent ones displaying similar gene syntenies to *B. subtilis* phage SPP1 (organization A) and *L. lactis* phage p2 (organization B). The other tail organizations are atypical, either lacking a conserved gene (no Dit protein could be identified in organization C) or encoding several tail proteins of unknown function (organizations D and E found in siphoviruses of genomes > 80 kb). It is noteworthy that CBM are widespread in Dit protein, Tal, and Tal/RBP found in *B. cereus* phages, giving further credence to the implication of carbohydrates in the adsorption of *B. cereus* phages. It was also experimentally shown that in the case of the siphophage Deep-Purple, the CBM found in the evoDit protein (i.e., Gp28) and in the Tal/RBP (i.e., Gp29) are both involved in the phage adsorption as GFP fused CBM were able to bind to *B. cereus* cells, although with different binding ranges [122]. Indeed, while the evoDit protein bound to sensitive strains but also some insensitive ones, the Tal/RBP could only bind to sensitive strains. This supports an accessory role for the evoDit protein of Deep-Purple, facilitating the phage adsorption while the binding of the Tal/RBP with its receptor may be the key interaction.

## 5. Conclusions

For many years, the potential of phages as antimicrobial agents has been overshadowed by their use as biological models to decipher the fundamentals of molecular biology. However, with the rise of antibiotic resistance, there is a renewed interest in studying phages with regard to their antibacterial potential. In addition, beyond their use in phage therapy, they are also promising in diverse fields and applications such as food safety and plant protection [173,174]. Considering that receptor mutation is, in many cases, the first cause of bacterial resistance to phages, it is now commonly admitted that phage therapy should rely on cocktails comprising several phages targeting different receptors to be effective and delay the appearance of bacterial resistances. Therefore, knowing the receptors involved in the adsorption of phages to pathogenic bacteria certainly facilitates the rational design of therapeutic cocktails [175]. In addition, in resistant bacteria, the modification of the targeted receptor may be associated with a reduction in virulence or an increase in sensitivity to antimicrobials [176]. Given that the apparition of phage resistance is unavoidable, phage therapy could also be employed, not only to kill pathogenic bacteria but also as a selective force to obtain surviving bacteria that can be treated more easily [11,176,177]. Nevertheless, isolating the right phages with appropriate features for therapy and biocontrol purposes (e.g., strictly virulent, non-transducing) and recognizing different receptors can be laborious. Hopefully, advances in RBP structure and synthetic biology now allow for modulating the phage host range by engineering RBP [92,178]. By relying on the modular structure of RBP, it is possible to exchange receptor-binding domains to obtain chimeric RBP with a shifted or extended host spectrum [92,179]. There is no doubt that the recent progress in protein structure prediction using AlphaFold will further improve the comprehension of host adhesion devices of phages and facilitate the engineering of phage RBP for therapeutic purposes [180,181].

## Figures and Tables

**Figure 1 viruses-15-00196-f001:**
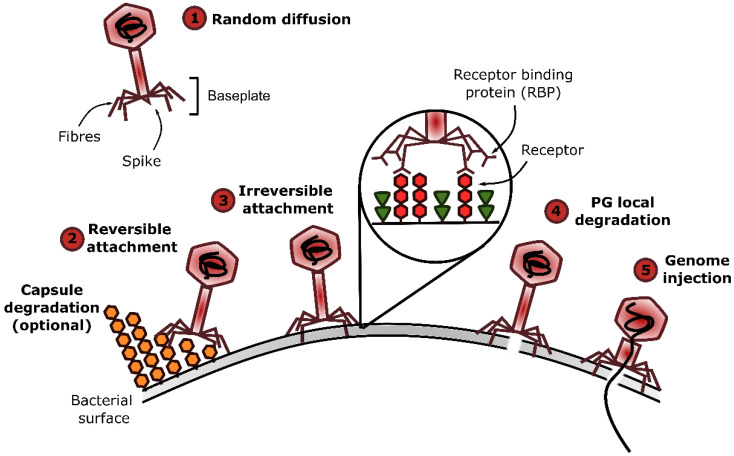
**Phage adsorption process and events leading to genome injection.** Phage adsorption to their host involves the interaction between Receptor Binding Proteins (RBP) located at the distal part of phage tails and receptors at the surface of the bacterial cell envelope. This process is divided into three steps: (1) random diffusion in the medium, (2) reversible attachment, and (3) irreversible attachment to the bacteria. The adsorption can be assisted by depolymerases cleaving capsular polysaccharides that can hamper receptor access. Following adsorption, virion-associated lysins may locally break down the PG (4) and, finally, conformational changes in the virion structure lead to genome injection (5). This figure was made with the open-source software Inkscape.

**Figure 2 viruses-15-00196-f002:**
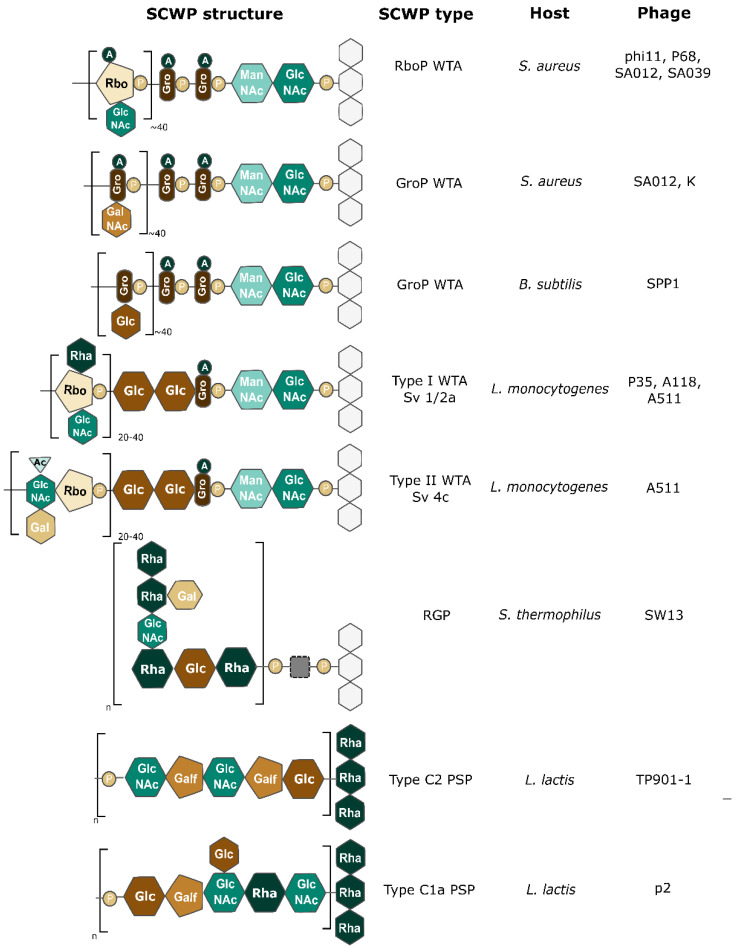
**Example of secondary cell wall polysaccharides acting as phage receptors.** Some SCWP structures are shown on the left with their corresponding SCWP types. The bacterial species in which each SCWP is found are shown on the right, together with one or several examples of phages using these SCWP as receptors. A: d-alanine; P: phosphate; Ac: acetylation. Glycosyl residues: Gal: galactose; GalNAc, *N*-acetylgalactosamine; Glc: glucose; Gro: glycerol; ManNAc, *N*-acetylmannosamine; Rbo, ribitol; Rha: rhamnose. SCWP: Secondary Cell Wall Polysaccharide; WTA: Wall Teichoic Acid; LTA: Lipoteichoic Acid; PSP: polysaccharide pellicle; RGP: Rhamnose Glucose polysaccharide. Gray squares indicate unclear linkage units. White hexagons represent PG. In *L. lactis* PSP are not directly linked to PG but rather to Rhamnose polysaccharides, which are attached to PG. Repeat units are indicated in brackets with numbers representing the number of repeats found in each SCWP, and n referring to an unknown number of repeats. This figure was made with Inkscape, with information from [40,66,70,71,72].

**Figure 3 viruses-15-00196-f003:**
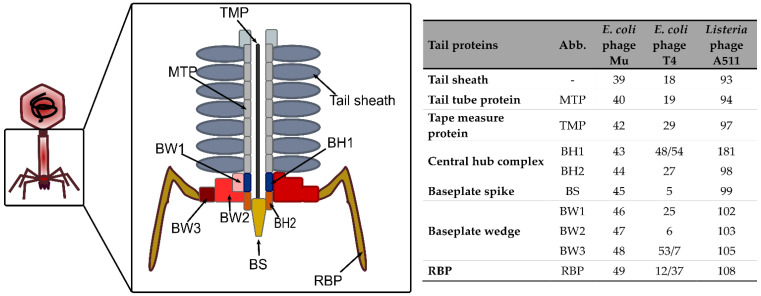
**Schematic representation of a myovirus contractile tail**. The tail of myoviruses infecting both Gram-positive and Gram-negative hosts harbor conserved proteins that are represented in this scheme. The complete baseplate wedge structure (formed by three types of proteins) is highlighted in red on the right side of the scheme. The table on the right gives the gene product correspondences for each baseplate protein between the two reference phages infecting *E. coli* (T4 and Mu) and the *L. monocytogenes* phage A511. TMP: Tape Measure Protein; BW: Baseplate Wedge protein; BH: Baseplate Hub protein; BS: Baseplate Spike; RBP: Receptor Binding Protein; TF: Tail Fiber; MTP: Major Tail Protein.

**Figure 4 viruses-15-00196-f004:**
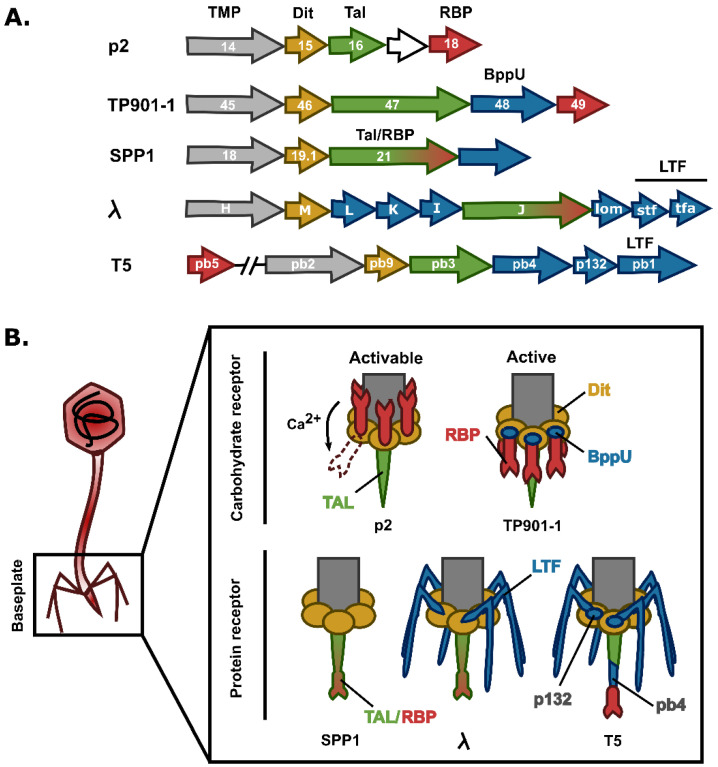
**Examples of baseplate organizations found in siphoviruses.** (**A**) Tail proteins are usually encoded between the TMP and the lysis cassette (not shown here). Gene identification is indicated in each arrow. TMP are shown in gray, Dit in yellow, Tal in green, RBP in red, other tail proteins in blue, and hypothetical proteins in white. (**B**) Schematic representation of p2, TP901-1, SPP1, λ, and T5 baseplates recognizing either carbohydrate or protein receptors. Color coding is that of the panel (**A**). TMP: Tape Measure Protein; Dit: Distal tail; Tal: Tail lysin; LTF: Long Tail Fiber; BppU: Upper Baseplate protein. This figure was made with Inkscape.

**Figure 5 viruses-15-00196-f005:**
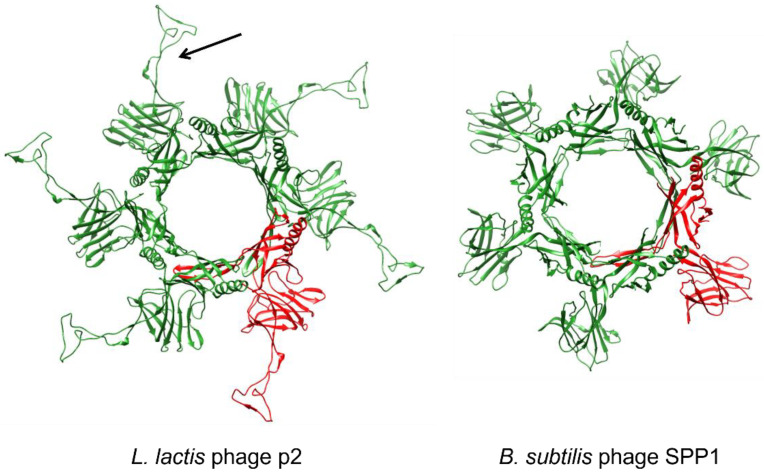
**Structure of Dit proteins of *L. lactis* phage p2 (left) and *B. subtilis* phage SPP1 (right).** Dit proteins form hexameric structures, and one monomer is highlighted in red. In p2 Dit protein, the “arm” structure is indicated by an arrow. Structures were retrieved from the Protein Data Bank (p2 Dit: PDB ID 4V5I and [119]; SPP1 Dit: PDB ID 2X8K and [118]).

**Figure 6 viruses-15-00196-f006:**
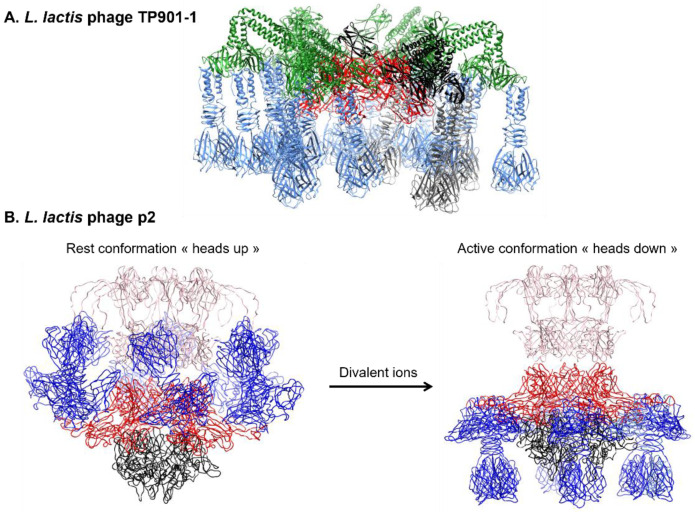
**Baseplate conformation of siphoviruses.** (**A**) Phage TP901-1 has a “ready-to-infect” baseplate, which is formed by a Dit hexamer (red) on which are anchored six trimeric BppU (green), each attaching three RBP trimer (blue). One tripod is formed by one BppU attaching three RBP and is highlighted in a different shade of black. The Tal trimer attaching at the bottom of the baseplate is missing. (**B**) In presence of divalent cation, phage p2 baseplate switches from a “rest” conformation, where RBP point upwards toward the capsid, to an “active” conformation where the RBP face down. The trimeric Tal (black) and the six RBP trimers (blue) are attached to a hexameric Dit (dark red). In phage p2, a second Dit hexamer is found in the baseplate (pink). Structures were retrieved from the Protein Data Bank (TP901-1 baseplate: PDB ID 4V96 and [91]; p2 baseplate rest: PDB ID 6ZJJ and [120]; p2 baseplate active: PDB ID 6ZIH and [120]).

**Figure 7 viruses-15-00196-f007:**
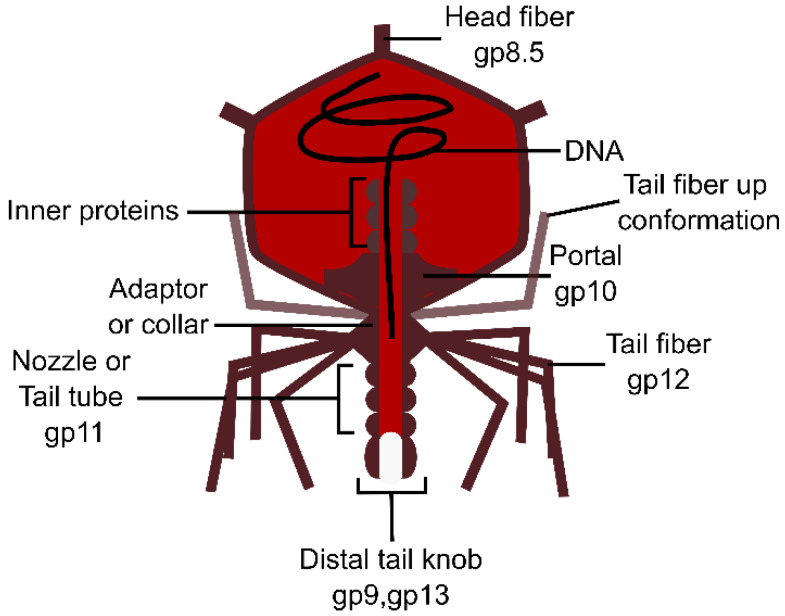
**Schematic representation of podovirus Φ29 infecting *B. subtilis*.** Gp numbers indicate the proteins involved in such structures in phage phi29. This figure was made with Inkscape.

**Figure 8 viruses-15-00196-f008:**
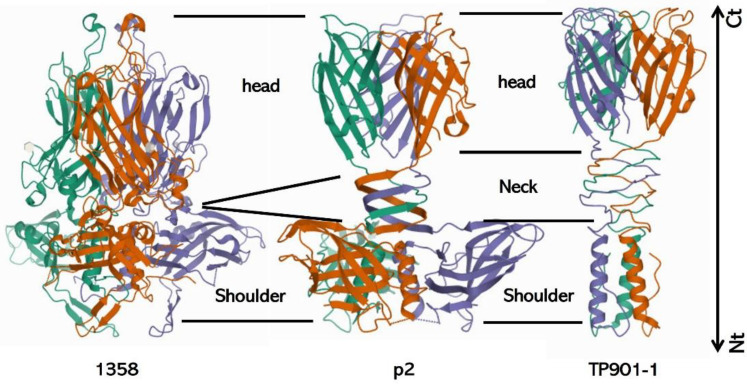
**Structure of lactococcal RBP.** Most lactococcal RBP harbor a three-domain organization with an N-terminal (Nt) shoulder domain, which attaches the RBP to the virion, a central connecting neck domain, and a C-terminal (Ct) head domain involved in receptor recognition. In some phages (e.g., 1358), the neck domain is absent. Each protein monomer is displayed in a different color. Structures were retrieved from the Protein Data Bank (1358: PDB ID 4RGA and [152]; p2: PDB ID 2BSD and [153]; TP901-1: PDB ID 3EJC and [134]).

**Figure 9 viruses-15-00196-f009:**
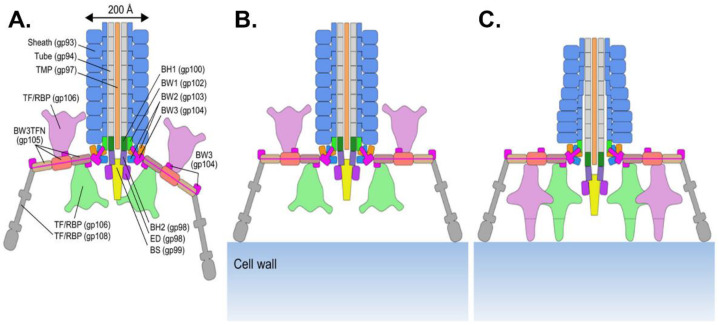
***Listeria* phage A511 baseplate structure and baseplate transformation upon adsorption**. (**A**) Phage A511 tail structure. The name of each tail protein is indicated with the related gene in parentheses. (**B**) A511 first interaction with the CW through binding of the RBP gp108. (**C**) Reorientation of the gp106 pyramidal structures and interaction with the CW. The distal part of the sheath begins to contract. BH: Baseplate Hub; RBP: Receptor Binding Protein; BW: Baseplate Wedge; BS: Baseplate Spike; TF: Tail fiber; TFN: Tail Fiber Network. Reprinted with permission from [99].

**Figure 10 viruses-15-00196-f010:**
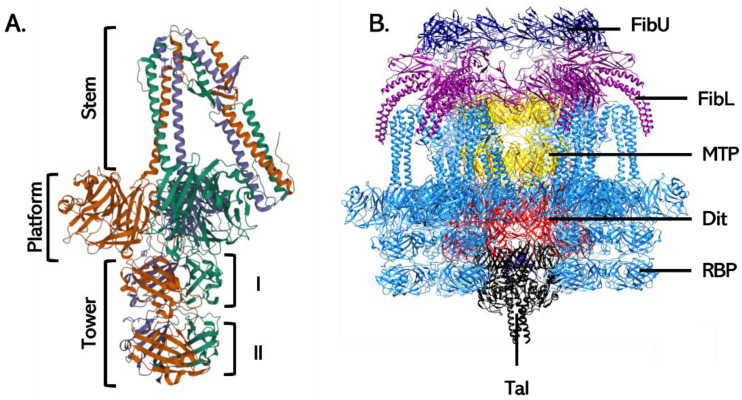
**Tail proteins constituting the baseplate of *S. aureus* siphoviruses.** (**A**) The RBP of ϕ11 (gp45) has a modular organization with three domains. N-terminal stem domain, central platform with the receptor binding site, and C-terminal tower domain consisting of two similar regions. (**B**) Phage 80α baseplate is particular in that it harbors, in addition to the RBP, two other tail fibers (FibL and FibU) that may be involved in adsorption. The Tal and FibU could not be completely modeled, and only the N-terminal part of the FibL are shown for clarity purposes. FibU: Upper tail fiber (dark blue); FibL: Lower tail fiber (purple); RBP: Receptor Binding Protein (blue); Tal: Tail lysin (black). MTP: Major Tail Protein (Yellow). Structures were retrieved from the Protein Data Bank (ϕ11 RBP: PDB 5EFV and [129]; 80α baseplate: 6V8I and [128]).

## Data Availability

No new data were created in this study. Data sharing is not applicable to this article.

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
