# Peer review of "Phage Adsorption to Gram-Positive Bacteria"

_viruses, 2023, doi:10.3390/v15010196_

Round 1

Reviewer 1 Report

This is a thoroughly researched and well organized review on the binding of phages to Gram-positive bacteria. There are some minor changes that need to be made.

Some reference titles are capitalized (for example, refs 69–90), and you'd be converted to lower case.

The text (and some figure legends) contain many species and genus names that need to italicized.

Reviewer 2 Report

As you point out in your introduction, the study of host recognition by phages is a key step in understanding and exploiting phage-bacteria interactions. It is a trendy and relevant topic for the reasons you well expose in the introduction and conclusion. It could perhaps even be remembered in the introduction that adsorption is common to all viruses.

The structure of the manuscript is well thought out and pedagogical, which is important for such a broad topic.

We should be aware that the ICTV has completely revised the classification of phages. In particular, the families myo-sipho-podoviridae no longer exist. The terms siphovirus ect. will probably continue to be used to describe the structure of phage particles, but it is important that publications during this transition periode are updated (L518).

It is conventional to italicise Latin: family names, genus and species (or even e.g. and e.i. ...).

L30-31, Phages are killers, but this property makes them regulators in the dynamics of microbial ecosystems. With regard to lysogeny, it is not a long quiet river since the spontaneous induction or not of prophages represents a "molecular time bomb", which may be costly to the lysogenic populations. Furthermore, prophages are able to modulate the cell surface of their host and thus influence their susceptibility to other phages, by encoding genes such as enzymes like glycosyltransferases or superinfection exclusion proteins. This aspect of lysogenic conversion might be mentioned somewhere in the manuscript. There is no reference.

L37-41, They are also referred to as primary and secondary receptors (or attachment factors).

L52-54, news on t5 https://doi.org/10.1073/pnas.2211672119 and https://doi.org/10.1101/2022.09.22.509047

Fig. 1, Specify a reference and/or the software used to make the figure.

L66, Or even the loss of the receptor.

L67, Not necessarily stochastic, more and more studies point out the importance of quorum sensing in particular.

L68, It would be appropriate to cite the phage BPP-1 (Liu et al., 2002).

L64-71 There are recent reviews dealing with this subject.

Fig. 2, Specify a reference and/or the software used to make the figure.

L80-87, In the way it is written, we are not sure that the IM is part of CW or not.

L126, In this part of the manuscript, the enzymes involved in the setting up of these structures could also be described (polymerases, flippases, sortases...).

L155, 2.2.2 not 1.2.2

L212, this part could talk about EPS in addition to cell wall polysaccharides and capsular polysaccharides ?

L252-260, No reference, especially necessary for the RBP which is a hotspot for evolution.

Fig. 4, Specify references and/or the software used to make the figure.

Fig. 5, Specify a reference and/or the software used to make the figure.

L434, auxiliary

L467, news on k1-like phage https://doi.org/10.1128/JVI.00920-21

L596, A511 not A551

L617-625, no reference

Fig. 11, no reference

L954, Bacteriol 2010 (one space is missing)

L968, Proc Natl Acad Sci USA 2016 (one space is missing)

L971, in Virol 2020 (one space is missing)

L1049, Appl Env Micro-1049 biol 2014 (a space is missing)

L1052, Microb Biotechnol 2020 (a space is missing)

L1108, (a space is missing)

L1113, (a space is missing)

L1117, (a space is missing)

L1155, (a space is missing)

I also noticed that the dates within the references are sometimes not in bold but I didn't systematically notice it.

Reviewer 3 Report

This review paper is devoted to a very important and fundamental process of initiation of bacteriophage infection of a bacterium. In general, the work is quite well written, but there are a number of remarks that I would like to draw attention to: 

1. Figure 1 is presented schematically and reflects mainly the process of penetration through the cell envelope of Gram-positive bacteria. However, Figure 2 is an illustration of cell wall variants without any links to the article on phage entry.  Moreover, why show the structure of the cell wall of a gram-negative bacterium, if the review is about gram-positive ones. Maybe it makes sense to imagine the cell wall of mycobacteria, for example, with an illustration of how phages penetrate through it?

2. Figure 3 is a description of various polysaccharide structures, which in itself is interesting in an article about the bacterial cell envelope, but in the context of an article on the absorption process, some kind of connection with phages is needed, for example, an indication of which phage receptor these polysaccharides are.

3. With all the abundance of information presented in the article, the main claim to work is the minimum information about the adsorption process. We learn from the work about baseplates, proteins involved in the work of the tail, examples about phages and their receptors, endolysins, but there is no information about what happens during adsorption.

4. There is no information about the classification of receptors in the article, there is no information about the classification of Receptor Binding Protein (RBP). That is, we have not gained new knowledge about the processes.

5. The situation is similar with endolysins and depolymerases. There is a mention of them, but their classification is completely absent, although they are in the literature. For example: Abdelrahman, F.; Easwaran, M.; Daramola, O.I.; Ragab, S.; Lynch, S.; Oduselu, T.J.; Khan, F.M.; Ayobami, A.; Adnan, F.; Torrents, E.; et al. Phage-Encoded Endolysins. Antibiotics 2021, 10, 124.

6. Figures 6, 7, and 9 are simply beautiful illustrations. What information for the reader in the context of phage adsorption?

7.Why the illustration concerning the T7 phage and its penetration through the periplasmic space of gram-negative bacteria (Figure 8) in the review about  gram-positive bacteria?

8. Despite the fact that I liked the work, it seems that adsorption is not the main topic. It is more related to the penetration of bacteriophages into the cell. But more than that, it resembles a description of various phages and their penetration through the bacterial cell envelope. In general, adsorption begins with the encounter of the phage to the cell and ends with irreversible binding to the receptor. That's it, the story about adsorption is over.

I would like a more accentuated look at what the authors see as the novelty of this review.

Round 2

Reviewer 3 Report

Perhaps I am unjustifiably picking on the work and asking questions that are not mentioned by other reviewers and which are difficult to answer, but these are the questions I expected answers from the article on phage adsorption.

The authors made corrections to the manuscript, but the main comments were not taken into account. The article has everything you want, but not the details of the adsorption process. Figure 1 shows the stages of the initial stages of phage infection, however, the stage of penetration through the capsule is omitted. Which immediately simplifies the picture of adsorption in the eyes of the reader. It would be interesting for the reader to know which enzymes are involved in the process in Gram-positive bacterial phages. Just mentioning that there are enzymes depolymerases is not enough. It is necessary to indicate where they are localized and how and when they are synthesized and how they participate in degradation. The same goes for reversible and irreversible binding. Initial recognition is carried out in relation to the same receptors as in irreversible binding? What determines the transition from a reversible to an irreversible state? Qualitative or quantitative process?

It seems to me that the article simply needs a table for the considered phages indicating (1) the enzyme involved in the degradation of the capsule (if any), (2) reversible binding receptors, (3) irreversible binding receptors (if not the same).

In Figure 1, a capsule penetration step needs to be added.

Figure 2 is clearly redundant. Its only need is if the authors would compare the receptors of gram-positive and gram-negative bacteria, but this does not happen. In its current state, it simply misleads the reader. For example, the figure does not show MDR pumps, which are phage receptors of Gram-negative bacteria. For example, the main MDR pump of E. coli (and not only) AcrAB-TolC is a phage receptor. I would recommend either deleting the figure or supplementing the text on the comparison of receptors and their binding mechanisms. Although the article is no need for this, except for saving an irrelevant drawing.
